# Screening a Targeted Panel of Genes by Next-Generation Sequencing Improves Risk Stratification in Real World Patients with Acute Myeloid Leukemia

**DOI:** 10.3390/cancers14133236

**Published:** 2022-06-30

**Authors:** Sónia Matos, Paulo Bernardo, Susana Esteves, Aida Botelho de Sousa, Marcos Lemos, Patrícia Ribeiro, Madalena Silva, Albertina Nunes, Joana Lobato, Maria de Jesus Frade, Maria Gomes da Silva, Sérgio Chacim, José Mariz, Graça Esteves, João Raposo, Ana Espadana, José Carda, Pedro Barbosa, Vânia Martins, Maria Carmo-Fonseca, Joana Desterro

**Affiliations:** 1GenoMed-Diagnósticos de Medicina Molecular SA, 1649-028 Lisboa, Portugal; smatos@medicina.ulisboa.pt (S.M.); vaniapmartins@medicina.ulisboa.pt (V.M.); 2Instituto de Medicina Molecular João Lobo Antunes, Faculdade de Medicina, Universidade de Lisboa, 1649-028 Lisboa, Portugal; paulo.bernardo@medicina.ulisboa.pt (P.B.); pedro.barbosa@medicina.ulisboa.pt (P.B.); 3Serviço de Hematologia Clínica, Hospital da Luz de Lisboa, 1500-650 Lisboa, Portugal; 4Unidade de Investigação Clínica, Instituto Português de Oncologia de Lisboa, Francisco Gentil, 1099-023 Lisboa, Portugal; sesteves@ipolisboa.min-saude.pt; 5Serviço de Hematologia, Centro Hospitalar Lisboa Central-Hospital de St. António dos Capuchos, 1150-315 Lisboa, Portugal; abs@netcabo.pt (A.B.d.S.); marcos.lemos@chlc.min-saude.pt (M.L.); patricia.m.ribeiro@chlc.min-saude.pt (P.R.); madalenaf.martins@chlc.min-saude.pt (M.S.); 6Serviço de Hematologia, Instituto Português de Oncologia de Lisboa, Francisco Gentil, 1099-023 Lisboa, Portugal; alnunes@ipolisboa.min-saude.pt (A.N.); jlobato@ipolisboa.min-saude.pt (J.L.); mjefrade@ipolisboa.min-saude.pt (M.d.J.F.); mgsilva@ipolisboa.min-saude.pt (M.G.d.S.); 7Serviço de Hematologia, Instituto Português de Oncologia do Porto, 4200-072 Porto, Portugal; sergio.chacim@ipoporto.min-saude.pt (S.C.); mariz@ipoporto.min-saude.pt (J.M.); 8Serviço de Hematologia e Transplantação de Medula, Centro Hospitalar Lisboa Norte-Hospital de Santa Maria, 1649-028 Lisboa, Portugal; graca.esteves@chln.min-saude.pt (G.E.); jraposo@netcabo.pt (J.R.); 9Serviço de Hematologia Clínica, Centro Hospitalar e Universitário de Coimbra, 3004-561 Coimbra, Portugal; ana.espadana@gmail.com (A.E.); jncarda@fmed.uc.pt (J.C.)

**Keywords:** AML, high-throughput sequencing, risk stratification

## Abstract

**Simple Summary:**

In this study, we prospectively analyzed a cohort of 268 newly diagnosed AML patients with the objective of assessing the clinical value of screening a targeted gene panel by next-generation sequencing (NGS). We found that access to NGS data refined risk assessment for 62 patients, corresponding to approximately 23% of the study population. We further compared clinical outcomes with prognostic stratification, and observed unexpected results associated with mutations in the *FLT3* gene, highlighting the need for further improvements in current risk classification criteria.

**Abstract:**

Although mutation profiling of defined genes is recommended for classification of acute myeloid leukemia (AML) patients, screening of targeted gene panels using next-generation sequencing (NGS) is not always routinely used as standard of care. The objective of this study was to prospectively assess whether extended molecular monitoring using NGS adds clinical value for risk assessment in real-world AML patients. We analyzed a cohort of 268 newly diagnosed AML patients. We compared the prognostic stratification of our study population according to the European LeukemiaNet recommendations, before and after the incorporation of the extended mutational profile information obtained by NGS. Without access to NGS data, 63 patients (23%) failed to be stratified into risk groups. After NGS data, only 27 patients (10%) failed risk stratification. Another 33 patients were re-classified as adverse-risk patients once the NGS data was incorporated. In total, access to NGS data refined risk assessment for 62 patients (23%). We further compared clinical outcomes with prognostic stratification, and observed unexpected outcomes associated with *FLT3* mutations. In conclusion, this study demonstrates the prognostic utility of screening AML patients for multiple gene mutations by NGS and underscores the need for further studies to refine the current risk classification criteria.

## 1. Introduction

Acute myeloid leukemia is a clonal malignant expansion of immature myeloid precursor cells that results in impaired hematopoiesis and bone marrow failure [1]. The finding that specific chromosomal structural variations were associated with significant differences in overall survival [2] led to the widespread adoption of cytogenetically defined risk stratification of patients [3]. However, approximately half of AML samples have a normal karyotype. More recently, advances in massively parallel sequencing unraveled an increasingly comprehensive landscape of genomic alterations in AML, providing additional biomarkers for disease classification and prognostic stratification [4,5,6].

Whole-genome sequencing of normal karyotype AML samples identified multiple mutations in numerous genes [7]. However, most of these mutations are also detected in healthy hematopoietic stem/progenitor cells and tend to accumulate during ageing [7]. Apparently, only a few cooperating driver mutations are needed to generate a malignant founding clone, and as the disease evolves, multiple competing clones that acquired additional mutations may coexist, contributing to progression and/or relapse [7].

A seminal study of 1540 AML patients described 11 subgroups of genomic alterations with different clinical outcomes [6]. Subsequently, mutation profiling data has been incorporated in the World Health Organization (WHO) 2016 classification of myeloid neoplasms [8], and in the European LeukemiaNet (ELN) 2017 recommendations for risk stratification of AML [9]. In particular, the ELN-2017 classification highlighted the prognostic relevance of gene mutations in patients with either normal or abnormal cytogenetics.

Diagnostic procedures currently used to classify AML include morphologic assessment of bone marrow specimens and blood smears, immunophenotyping by flow cytometry, cytogenetic testing, and molecular genetics screening [10]. For molecular testing, the gold standard technologies are polymerase chain reaction (PCR) and high-throughput sequencing (also known as next-generation sequencing, NGS). High-throughput sequencing offers many advantages over previous methods, namely a high sensitivity and capacity for massive parallelization. In a single sequencing run, a large number of genomic loci can be screened in multiple samples [10]. Although screening of targeted gene panels by NGS is broadly recommended for AML diagnosis [8,10], high-throughput sequencing data is not yet routinely available in all clinical centers.

In this study, we evaluated a cohort of 268 patients newly diagnosed with AML in Portugal, where NGS is not routinely used at diagnosis, to investigate whether extended molecular monitoring by NGS could add clinical value for risk assessment.

## 2. Materials and Methods

Patient cohort: Although eligible patients comprised adults with a diagnosis of AML (initial, relapsed, or refractory), the final cohort analyzed comprised 268 newly diagnosed patients and only one patient with relapsed disease. Biological samples were collected as part of routine clinical practice in five health care centers across Portugal. All patients were treated according to the centers’ standard protocols without knowledge of the NGS results. All patients gave informed consent for genetic analysis according to the Declaration of Helsinki. This study was approved by the institutional ethical review board of each center.

Mutation profiling: AML diagnostic bone marrow aspirates or peripheral blood samples were collected. Mononuclear cells were enriched using Ficoll density gradient centrifugation. Genomic DNA was extracted using the QIAamp DNA Blood Mini Kit (Qiagen, Hilden, Germany) according to the manufacturer’s instructions. The TruSight Myeloid Sequencing Panel (Illumina, San Diego, USA) was used, starting from 50 ng of genomic DNA and according to the manufacturer’s instructions. The entire coding sequences of 15 genes and exonic hotspots of an additional 39 were accessed. The resulting libraries were pooled and sequenced on an Illumina MiSeq instrument (Illumina, San Diego, USA) using 2 × 150 bp paired-end reads. The sequences were aligned to the human genome reference NCBI37/hg19. Data was analyzed with MiSeq Reporter software, version 2.6.2.3 (Illumina, San Diego, USA). Variant Studio software, version 3.0 (Illumina, San Diego, USA) was used for variant calling and annotation, with a limit minimum of the Variant Allelic Fraction of 5% (VAF ≥ 5%) and minimum coverage of 500×. Nonsynonymous variants in coding regions were classified based on their deduced consequences on the amino acid level. Nonsense and frameshift mutations with a premature stop codon were considered pathogenic. Missense variants, in-frame insertions/deletions, and splice site variants were classified based on the available data on Catalogue Of Somatic Mutations In Cancer (COSMIC, version 90), OncoKB database, MyCancer Genome database, National Center for Biotechnology Information Short Genetic Variations database (dbSNP, version 153), NHLBI 1000 genomes cohort, their location relative to known mutations in the gene, and their predicted functional consequences using in silico predictions such as Mutation Taster algorithm.

Statistics: We performed a descriptive analysis using absolute and relative frequencies for categorical variables and the mean and standard deviation or median and minimum and maximum, in the case of an asymmetric distribution, for quantitative variables. We compared the genetic mutation distribution by age group (<60 vs. ≥60 years old) using Pearson’s Chi-squared test or Fisher’s exact test, as appropriate. Fisher’s exact test was used for assessing pairwise association and exclusivity between driver genes and complex karyotype, with adjustment for multiple testing using the Benjamini–Hochberg method. Overall survival was defined as the time from diagnosis until death from any cause. Median follow-up was calculated using the reverse Kaplan–Meier method. Overall survival was analyzed using the Kaplan–Meier method. The log-rank test was used for ELN-2017 genetic risk group comparison. We also evaluated the independent impact of the genetic variants associated with myelodysplasia and those proposed for refinement of ELN-2017 risk classification on overall survival using the Cox proportional hazards model adjusted for the ELN-2017 risk group and age category. Schoenfeld residuals were used to check the proportionality assumption. All tests were two-sided, and we considered a significance level of 0.05. The data was analyzed using R (https://www.r-project.org/ accessed on 25 May 2022).

## 3. Results

### 3.1. Patient Characteristics and Treatments

This study enrolled 301 patients between 2016 and 2019. Patients were not consecutively recruited, and no selection criteria were used. AML diagnosis was not confirmed in 23 patients, and these were excluded. Another nine patients were further excluded due to technical issues during mutation profiling by NGS. Table 1 displays the demographic and laboratory baseline characteristics of the 269 patients that comprise the study population. Among the analyzed patients, 268 were newly diagnosed, and only one had relapsed disease. The median age was 61 years (min-max, 21–100 y) and the gender distribution was equilibrated. More than half of the patients (54%) were older than 60 years. The majority of the patients (88.5%) had a good ECOG performance status. Leukocyte counts ranged from 0.2 to 339 × 10^9^/L (median, 17 × 10^9^/L). Hemoglobin levels ranged from 3.7 to 14.7 × 10^9^ g/dL (median, 8 g/dL), and platelet counts ranged from 2 to 622 × 10^9^/L, with a median of 47 × 10^9^/L. Most patients had high blast counts at presentation (median 63%). However, two patients showed blast counts lower than 20%, one with myeloid sarcoma (blast count 10%) and the other with relapsed disease (blast count 14%).

Cytogenetic studies were not available for all patients. From the 178 patients analyzed by conventional cytogenetics, 51% had recurrent genetic abnormalities according to the World Health Organization 2016 classification criteria [8]. Moreover, 68 patients (25,8%) had AML with nucleophosmin (NPM1) mutation and 54 patients (20%) had AML with myelodysplasia-related changes (Table 2). In the population aged ≥60 years, the frequency of AML with myelodysplasia-related changes was higher than in the younger population (26.7% and 12.2%, respectively). Therapy-related AML was slightly higher in the younger population (13% vs. 7.5%).

The best therapeutic option for each patient was defined based on age, co-morbidities, and ECOG performance status. Notably, NGS data was obtained prospectively and did not influence the clinical management of patients. Most patients (66.5%) received standard induction intensive chemotherapy with cytarabine for 7 days and mitoxantrone or anthracyclines (idarubicin or doxorubicin). Alternative treatment schemes included hypomethylating agents (36 patients, 14.1%) and best supportive care (32 patients, 11.9%). From 44 patients with *FLT3*-ITD mutations, 7 were treated with Midostaurin, and only 2 in combination with induction intensive chemotherapy. This low number is in part explained because many patients were recruited before national authorities approved the use of FLT3 inhibitors. The group of patients treated with best supportive care had a median age of 77 years, and the only 2 patients younger than 60 years had relevant co-morbidities (one had metastatic ovarian cancer and the other had a past history of ischemic stroke and ECOG PS 3). Nine patients (3.3%) died before starting AML treatment (Table 2). Overall, complete remission (CR) was 69% (105/179) after AML induction treatment. In total, 44 patients were refractory, and 14/179 died during induction treatment (corresponding to a 7.9% induction-related mortality). Allogeneic transplant was performed in 35 out of 179 (20%) patients treated with intensive induction chemotherapy (corresponding to 13% of the total study population). None of the patients treated with hypomethylating agents proceeded to transplant.

With a median follow-up of 27.6 months (95% CI 22.4–31.3 months), 189 patients died. The median overall survival of the study cohort was 8.47 months (95% CI 6.2 to 11.7 months). The overall survival at 2 years was 29% (95% CI 24% to 36%). As shown in Figure 1, the use of intensive chemotherapy was associated with a survival advantage compared with low-intensive chemotherapy (AZA/LDAC) and best supportive care (BSC). A median overall survival of 14.5 months (95% CI 10.7–22.9 months) was observed with intensive-chemotherapy-treated patients, 8.5 months (95% CI 5.9 –13.6 months) in the low-intensive chemotherapy group and 1.1 months (95% CI 0.8–2.4 months) in patients only treated with best supportive care.

### 3.2. Mutation Profiling

The TruSight Myeloid Sequencing Panel used in this study is a targeted, multiplexed amplicon-based approach that targets 54 genes associated with myeloid disease, covering full exonic regions of 15 genes and exonic hotspots of an additional 39 genes. We only considered exonic, splice site, and non-synonymous variants and we restricted the analysis to variants described as pathogenic or potentially pathogenic.

We grouped the genes that were most frequently mutated into categories according to their biological function in the cell (Figure 2a): nucleophosmin (*NPM1*); DNA methylation (*DNMT3A*, *TET2*, *IDH1*, *IDH2*); cell signaling (*FLT3*, *NRAS*, *PTPN11*, *KRAS*, *KIT*, *CBL*, *KIT*, *JAK2*, *MPL*, *NOTCH1*, *CSF3R*, *BRAF*, *ABL1*, *HRAS*); epigenetic modifiers (*ASXL1*, *BCOR*, *EZH2*, *BCORL1*); tumor suppressors (*TP53*, *WT1*, *PHF6*); transcription factors (*RUNX1*, *GATA2*, *CEBPa*, *ETV6*, *CUX1*, *SETBP1*, *IKZF1*, *GATA1*); RNA splicing (*SRSF2*, *U2AF1*, *SF3B1*, *ZRSR2*); and cohesin complex (*STAG2*, *RAD21*, *SMC1A*, *SMC3*). Next, using the European LeukemiaNet (ELN) 2017 classification criteria [8], we stratified the patients harboring each mutated gene as favorable, intermediate, or adverse risk (Figure 2a).

For risk stratification, we used both high-throughput sequencing data and alternative molecular tests performed in each clinical center. For example, *FLT3*-ITD mutations were screened by PCR followed by capillary electrophoresis. This method detected 44 positive samples, whereas high-throughput sequencing detected only 21. Such a discrepancy was expected because the NGS technology used in this study can identify small but not medium and large internal tandem duplications [11]. Similarly, mutations in the *NPM1* gene, the most frequent being a 4 base pair insertion of TCTG, were identified by NGS in 58 samples and by PCR followed by capillary electrophoresis [12] in 76 samples. In contrast, more than 50% of the samples harboring the *FLT3*-TKD mutation, which consists of a point mutation in the TK domain [13], were only detected by high-throughput sequencing.

In our molecular analysis, the most frequently mutated gene was *NPM1* (76 patients, 28.3%), followed by *DNMT3A* (67 patients, 24.9%) and *FLT3* (57 patients, 21.2%). The *IDH2*, *IDH1*, *TP53*, *ASXL1*, and *RUNX1* mutations occurred in 36 (13.5%), 28 (10.4%), 27 (10.0%), 27 (10.0%), and 25 (9.3%) patients, respectively. Seven genes (*NPM1*, *DNMT3A*, *FLT3*, *TP53*, *ASXL1*, *IDH2*, and *IDH1*) had a mutation prevalence ≥10%.

According to the ELN-2017 classification criteria, mutations in the *NPM1* gene characterized the favorable and intermediate risk category, whereas mutations in the *ASXL1*, *RUNX1*, and *TP53* genes were associated with adverse risk. Most other variants presented a similar distribution across the different risk categories (Figure 2a).

We found a significant co-occurrence of mutations in the *NPM1* and *FLT3* or *DNMT3A* genes. Additionally, mutations in the *FLT3* gene co-occurred with mutations in the *DNMT3A* gene, mutations in the *BCOR* gene co-occurred with mutations in the *BCORL* gene, and mutations in the *TP53* gene co-occurred with complex karyotype alterations (Figure 2b). In contrast, mutually exclusive mutations were found in *NPM1* and *TP53*; *NPM1* and *RUNX1*; and *AXL1* and *DNMT3A* (Figure 2b).

### 3.3. Mutational Landscape According to Age

We identified at least one pathogenic variant in 249 (92%) patients, and more than half of the patients (149/269) had at least 3 pathogenic variants. The median number of pathogenic variants identified per patient was three. Compared to younger patients, elderly patients tended to have more mutations (Figure 3a). The frequency of mutated genes in younger (<60 years) and elderly (≥60 years) patients is shown in Figure 3b. In the elderly group, the more frequently mutated genes were *DNMT3A* (26.7%), *NPM1* (26.7%), *TET2* (20.5%), *ASXL1* (18.5%), *FLT3* (17.8%), and *TP53* (15.1%). In younger patients, the more frequently mutated genes were *NPM1* (29.4%), *FLT3* (24.5%), and *DNMT3A* (22.2%).

The presence of mutations in at least one gene associated with myelodysplasia (i.e., *SRSF2*, *SF3B1*, *U2AF1*, *ZRSR2*, *ASXL1*, *EZH2*, *BCOR*, or *STAG2*) [14] was significantly more frequent in older patients (Table 3). ELN-2017 high-risk mutations in the *ASXL1*, *RUNX1*, and *TP53* genes were also more frequently identified in older patients (Table 3).

No significant difference between age groups was detected for the frequency of gene mutations that were recently proposed as a refinement of the ELN-2017 criteria [15]. These included mutations in the *BCOR, STEB1* and *IDH2* genes (Table 3) that were associated with intermediate risk [15], and mutations in the *DNMT3A* and *ZRSR2* genes, and the co-occurrence of mutations in the *NPM1* and *WT1* genes (Table 3) were associated with adverse risk [15].

### 3.4. Survival Analysis

After patient stratification according to the ELN-2017 criteria, we estimated an overall survival of 14.3 months (95% CI 11.37-NA) in the favorable group, 12.2 months (95% CI 5.7–31.9) in the intermediate group, and 5.6 months (95% CI 3.17–9.8) in the adverse group (Figure 4a).

Due to the lack of information on the *FLT3*-ITD allelic burden [8], we adapted the ELN-2017 criteria and classified patients with a co-occurrence of *FLT3*-ITD and *NPM1* mutations as favorable/intermediate risk, and patients with *FLT3*-ITD mutations in the absence of *NPM1* mutations as intermediate/adverse risk (Figure 4). We expected that the overall survival in the favorable/intermediate group would be better, compared to the intermediate group. However, we found a median overall survival of 6.12 months (95% CI 1.87-NA), which is half the median survival time observed in the intermediate group. Our favorable/intermediate group comprised 26 patients with a median age of 58 years (min-max, 23–83), and the majority (96%) had mutations in additional relevant genes. Namely, 13 (50%) patients had mutations in *DNMT3A*, 8 (30%) had either *IDH1/2* mutations, and 5 (19%) had a combination of mutations in *NPM1*, *FLT3 DNMT3A*, and *IDH1/2*. In the latter group, 2 patients died before any treatment, and of the 19 treated with intensive induction chemotherapy, 3 died during induction. This corresponds to an induction-related mortality two-fold higher than that observed in the total population of intensive-treated patients (7.9% vs. 15.7%). When compared to the intermediate group, the favorable/intermediate group had statistically significantly more patients with *DNMT3A* mutations (50% vs. 22%; *p* = 0.013). Mutations in *IDH1/2* were also more frequent in the latter group; however, this did not reach statistical significance (30% vs. 20%; *p* = 0.29). Neither mutations in myelodysplastic-related genes nor mutations in genes proposed as a refinement of the ELN-2017 criteria had significant variation between the two groups of patients. In addition, the median number of variants in the favorable/intermediate group was statistically higher than in the intermediate group (3.0 vs. 1.0; *p* < 0.001). Both the higher prevalence of *DNMT3* mutations and additional co-occurring mutations could contribute to the unfavorable outcome of these patients.

We further expected that the overall survival in the intermediate/adverse group would be worse, compared to the intermediate group. In contrast, the overall survival in the intermediate/adverse group was 30.30 months (95% CI 8.47-NA), which is much longer than the median survival time observed in the favorable group. No deaths were observed during induction of the nine patients treated with intensive chemotherapy nor during treatment of three patients with FLT3 inhibitor. Our intermediate/adverse group comprised 11 patients with an average age of 56 years (min-max, 26–70). Three patients had mutations only in the *FLT3* gene, five patients had a concurrent pathogenic variant in *DNMT3A* (with or without additional mutations), and three patients had a concurrent pathogenic variant in genes other than *DNMT3A*. Within this group, the outcome was worse for the five patients with combined *FLT3*-ITD and *DNMT3A* mutations.

The data shown in Figure 4 clearly reveals that *FLT3*-mutated AML is a very heterogeneous group with variable outcomes irrespective of the treatment options (Figure 4b–d) and age (Figure 4e,f). However, the small cohort of AML patients analyzed in this study limits the accuracy of potential genotype–phenotype correlations.

Next, we performed a multivariable analysis of selected gene mutations adjusted for age at diagnosis (categorized as either <60 or ≥60 years old) and the ELN-2017 risk group. We analyzed mutations in myelodysplastic-related genes [14], and gene mutations proposed as a refinement of the ELN-2017 criteria [15]. As shown in Table 4, mutations in myelodysplastic-associated genes were not independent predictors of overall survival (HR 0.70; 95% CI 0.49–1.00, *p*= 0.051). Similarly, none of the mutations proposed as a refinement of the ELN-2017 criteria had an independent prognostic value for overall survival. Only *TP53* mutations represented an independent predictor of adverse overall survival (HR 2.96, 95% CI 1.81–4.84, *p* < 0.001).

Finally, we classified our study population according to the ELN-2017 criteria before and after incorporation of the extended mutational profile information obtained by NGS. As shown in Figure 5a, the number of patients without criteria for risk assessment before access to NGS data was 63 (23%). After incorporation of NGS data, 33 patients were newly classified as adverse risk, 2 patients were newly classified as favorable, and 1 patient was newly classified as favorable/intermediate due to the presence of an *FLT3*-ITD mutation. Only 27 patients (10%) failed to be stratified into risk groups after the incorporation of NGS data, and this was mainly due to the absence of cytogenetic information in this group of patients. An additional group of patients that were previously classified as favorable (3 patients), favorable/intermediate (2 patients), intermediate (20 patients), and intermediate/adverse (1 patient) risk before access to NGS data were reclassified as adverse-risk patients once the NGS data was incorporated (Figure 5a).

The Kaplan–Meier curves for the overall survival of patients classified as adverse risk after the incorporation of NGS data confirmed the dismal outcome (Figure 5b). Indeed, the overall survival of patients reclassified as adverse risk due to the identification of mutations in high-risk genes (*ASXL1*, *RUNX1*, and *TP53*) was similar to the overall survival of patients classified as adverse risk irrespective of the NGS data (OS 5.3. months, 95% CI 3.17–11.0, and OS 6.17 months, 95% CI 2.67–11.4, respectively).

## 4. Discussion

This study demonstrates that the use of a myeloid cancer gene panel and high-throughput sequencing adds clinical value for risk assessment in real-world newly diagnosed AML patients. We detected a clear benefit for 62 patients, corresponding to approximately 23% of the study population. Without access to NGS data, these patients were either misclassified or failed risk classification according to the ELN-2017 recommendations. Thus, our work provides valuable quantitative information for the ongoing debate on the cost-effectiveness of including extended mutation profiling by NGS in the routine diagnosis of AML patients.

As previously reported, we identified the *NPM1*, *FLT3*, and *DNMT3A* genes as the most frequently mutated [4,5,6]. In agreement with previous studies, we found a higher burden of adverse-risk mutations in elderly patients [14,16]. Furthermore, we observed that elderly patients had a significantly higher prevalence of myelodysplastic-associated gene mutations. In this regard, there is prior evidence suggesting that a previous myelodysplastic syndrome associates with more adverse outcomes [14,17].

We confirmed that mutations in three genes (*ASXL1*, *RUNX1*, and *TP53*), considered as adverse genetic markers, are indeed associated with a dismal outcome (Figure 5b). However, we found heterogeneous outcomes associated with mutations in the *FLT3* gene (Figure 4). *FLT3* is a transmembrane receptor with ligand-activated tyrosine kinase (TK) activity that plays an important role in the early stages of both myeloid and lymphoid lineage development [18]. Mutations in the *FLT3* gene are found in approximately 30% of newly diagnosed AML cases [13]. The most common alteration (~25%) consists of in-frame duplication of small sequences, ranging from 3 to >400 base pairs (*FLT3*-ITD). Less frequently (7–10%), point mutations occur in the TK domain (*FLT3*-TKD). Both *FLT3*-ITD and *FLT3*-TKD mutations lead to constitutively active *FLT3* kinase activity, resulting in the proliferation and survival of AML [13]. In particular, *FLT3*-ITD is one of the three most common driver mutations in AML [19], and a target for novel therapeutic approaches [13].

Although patients with *FLT3*-ITD mutations tend to have a shorter overall and relapse-free survival compared with patients without the mutation [20], the mutant-to-wild-type allelic ratio, insertion site, ITD length, karyotype, and the simultaneous presence of a mutation in the *NPM1* gene were reported to influence the prognostic utility of *FLT3*-ITD in AML patients [20,21]. Based on this evidence, ELN-2017 recommends that, along with *FLT3*-ITD screening, the mutant-to-wild-type allelic ratio should be determined. However, estimation of the *FLT3*-ITD allelic burden is not yet part of the standard testing in every clinical practice [13].

According to the ELN-2017 criteria, *NPM1*-mutated AML with a low ratio of mutant to wild-type *FLT3*-ITD alleles is considered as a favorable prognostic subgroup, similar to AML with absent *FLT3*-ITD mutation. In contrast, AML patients with a high ratio of mutant to wild-type *FLT3*-ITD alleles tend to have reduced complete remission rates, with poor survival and relapse. In this study, because *FLT3*-ITD allelic ratio data was not available, we grouped patients with a co-occurrence of *FLT3*-ITD and *NPM1* mutations. We expected that these patients had favorable/intermediate risk. However, their outcome was less favorable than that observed in the intermediate group (Figure 4). Further analysis revealed that the majority (96%) of these patients had mutations in the *FLT3*, *NPM1*, and additional genes, including *DNMT3A* and *IDH1/2*. In agreement with these findings, recent studies reported poor prognosis associated with concurrent mutations in the *FLT3*, *NPM1*, *DNMT3A*, and *IDH1/2* genes [22,23,24]. To date, the ELN-2017 risk score only includes the co-occurrence of mutations in the *NPM1* and *FLT3* genes [9]. However, it is likely that additional concurrent mutations will be assigned with prognostic value in future risk stratification systems.

We further identified a group of patients with *FLT3*-ITD mutations but no alterations in *NPM1*. We expected that these patients had intermediate/adverse risk, but in fact, their outcome was better than that observed in the favorable group (Figure 4). Nevertheless, within this group, the outcome was worse for those patients with combined *FLT3*-ITD and *DNMT3A* mutations, reinforcing the view that the concurrent alteration of the two genes has a negative impact on prognosis [22,23,24].

Although mutations in genes such as *NPM1* and *CEBPA* are considered a favorable prognostic marker, and *TP53* mutations represent an adverse prognostic marker, the vast majority of patients carry mutations in multiple genes. Moreover, patients may harbor distinct mutations in the same gene. How prognosis is influenced by combinatorial patterns of different types of mutations is poorly understood. In this regard, further refinements of the ELN-2017 classification have been proposed by the inclusion of additional combinations of genetic markers [15,25]. However, the development of algorithms that integrate genetic and other risk factors such as age, comorbidities, and performance status to guide individualized AML clinical decisions remains a critical unmet need. Future large-scale prospective real-word studies with properly registered data and guaranteed accessibility to clinical outcomes will contribute towards this ambitious goal by providing representative evidence from routine practice about the clinical outcomes of patients.

Another challenge is to understand how the AML genetic diversity captured by NGS-based approaches can be used to guide treatment in clinical practice. In this regard, a decision tool based on seven mutated genes has been proposed to identify fit elderly patients for intensive induction chemotherapy [26]. A detailed census of gene mutations is also likely to become increasingly important for prediction of the response to emerging targeted treatments such as the IDH inhibitors enasidinib and ivosidenib, which have been approved by FDA for AML patients with *IDH2* and *IDH1* mutations [11]. Finally, NGS-based approaches enable monitoring molecular minimal residual disease, but the prognostic value of persistent mutations remains to be established [27].

## 5. Conclusions

This real-world study demonstrates the prognostic utility of NGS in AML, and highlights the importance of profiling multiple genes to stratify patients with FLT3 mutations.

## Figures and Tables

**Figure 1 cancers-14-03236-f001:**
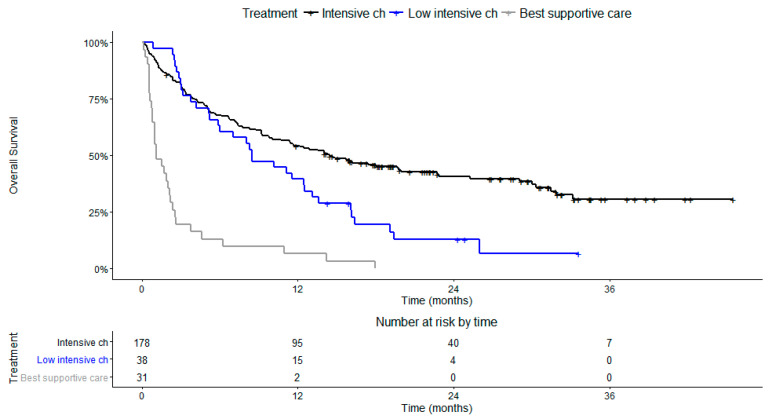
Kaplan–Meier curves for overall survival according to treatment. Intensive chemotherapy; low-intensive chemotherapy, including azacytidine and LDAC; and best supportive care (BSC).

**Figure 2 cancers-14-03236-f002:**
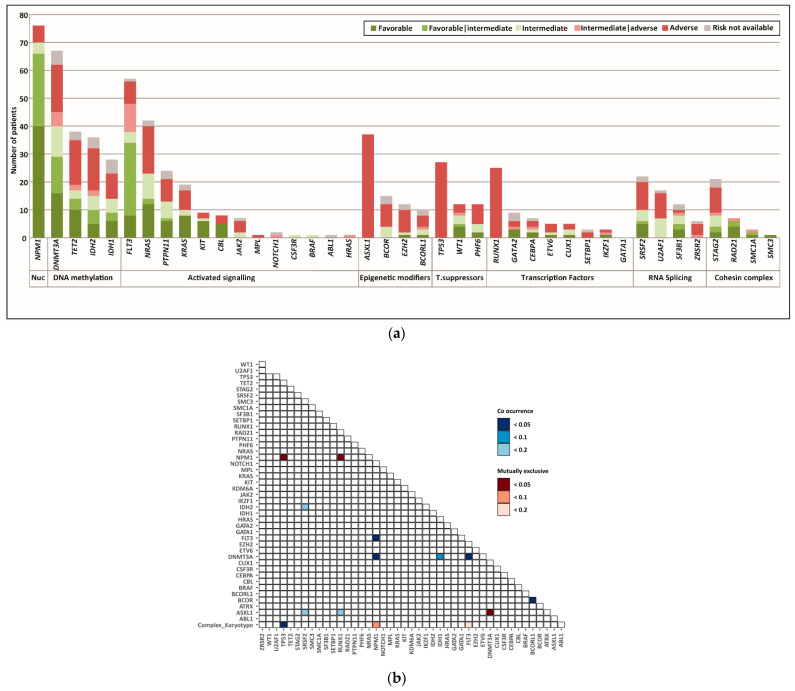
Mutational landscape of the study population. (**a**) Mutated genes were grouped into functional categories. Each bar represents patients harboring a mutation in the indicated gene. The colors in each bar indicate the risk group according to the ELN classification. Because the *FLT3*-ITD allelic burden was not available, patients containing both *FLT3*-ITD and *NPM1* mutations were considered favorable|intermediate risk, whereas patients containing *FLT3*-ITD but no *NPM1* mutation were considered intermediate|adverse risk. (**b**) Pairwise associations among somatic mutations and complex karyotype alterations. Note that mutational profiling data was available for 269 patients, but karyoptype analysis was only available for 180 patients. Pairings with an adjusted q-value < 0.05 were considered to be significant. Pairings with more lenient q-value thresholds (0.1 and 0.2) are also presented. Blue colors indicate a positive association. Red colors indicate a negative association.

**Figure 3 cancers-14-03236-f003:**
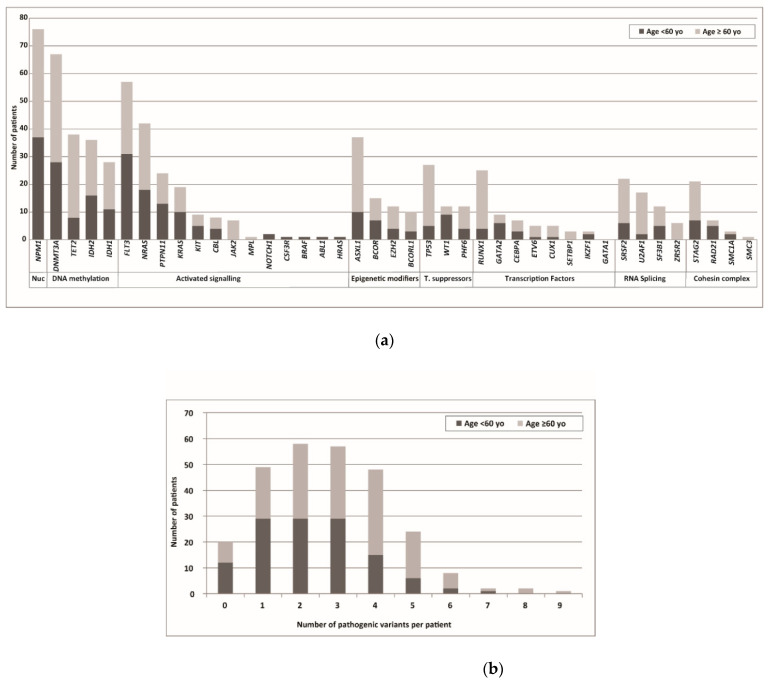
Mutational landscape according to age. (**a**) Mutated genes were grouped into functional categories as in Figure 2. Each bar represents patients harboring a mutation in the indicated gene according to age, in younger (<60 years) and elderly (≥60 years) patients. (**b**) Number of pathogenic variants identified per sample in younger (<60 years) and elderly (≥60 years) patients.

**Figure 4 cancers-14-03236-f004:**
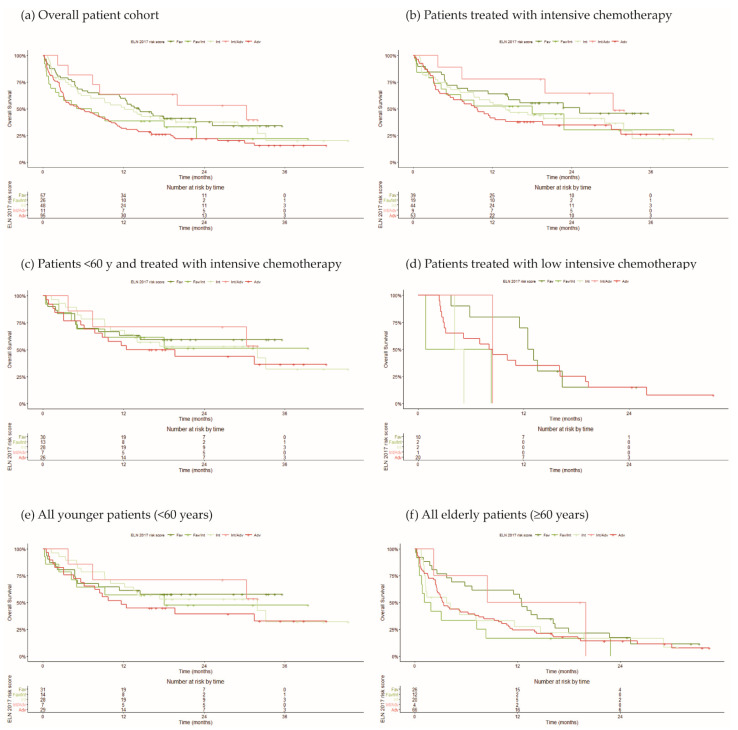
Kaplan–Meier curves for overall survival. Patients were stratified into favorable, intermediate, and adverse risk groups according to the ELN-2017 criteria. Due to the lack of information on the *FLT3*-ITD allelic burden two additional groups were considered: favorable|intermediate and intermediate|adverse. (**a**) Overall patient cohort. (**b**) Patients treated with intensive chemotherapy. (**c**) Patients <60 y and treated with intensive chemotherapy. (**d**) Patients treated with low-intensive chemotherapy. According to age, in younger (<60 years) (**e**) and elderly (≥60 years) patients (**f**).

**Figure 5 cancers-14-03236-f005:**
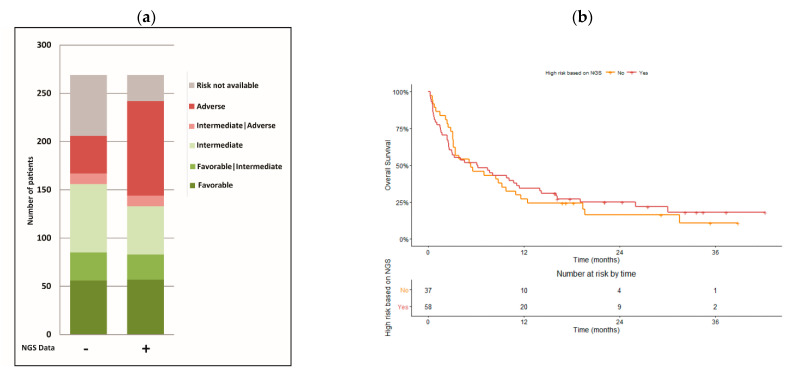
Extended mutation profiling improves risk stratification. (**a**) Risk stratification of patients was performed according to the ELN-2017 criteria, before (−) and after (+) incorporating the extended mutation profiling data obtained by high-throughput sequencing (NGS). Due to the lack of information on the *FLT3*-ITD allelic burden, two additional groups were considered: favorable|intermediate and intermediate|adverse. (**b**) Kaplan–Meier curves for the overall survival of patients classified as adverse risk irrespective of data obtained by high-throughput sequencing (NGS no), and patients reclassified as adverse risk because high-throughput sequencing identified mutations in the *ASXL1*, *RUNX1*, and *TP53* genes (NGS yes).

**Table 1 cancers-14-03236-t001:** Demographic and laboratory baseline characteristics of the study population.

Variable	Total Cohort, *n* = 269	Younger Patients (<60 y), *n* = 123	Older Patients (≥60 y), *n* = 146
**Age** (**y**)			
Median	61	48	71
Min-max	21–100	21–59	60–100
**Female gender, *n*** (**%**)	135 (50)	64 (52)	71(49)
**ECOG Performance status, *n*** (**%**)			
0–1	238 (88.5)	122 (99.2)	116 (79.5)
2	26 (9.7)	0	26 (17.8)
3–4	5 (1.8)	1 (0.8)	4 (2.7)
**WBC** (**count ×10^9^/L**)			
Median	17.0	19.0	12.0
Min-max	0.2–339	0.2–339	0.4–283
**Hemoglobin** (**g/dL**)			
Median	8.0	9.0	8.0
Min-max	3.7–14.7	3.7–14.7	4.0–13.5
**Platelet** (**count ×10^9^/L**)			
Median	47.0	48.0	45.0
Min-max	2.0–622	2.0–622	4.0–500
**Bone marrow blasts** (**%**)			
Median	63	69	59
Min-max *	10.0–100	14.0–100	10.0–98

* The asterisk highlights that in the indicated group, a patient had myeloid sarcoma and 10% of blasts while another patient had relapsed AML and 14% of blasts. ECOG: Eastern Cooperative Oncology Group; WBC: White Blood Cell.

**Table 2 cancers-14-03236-t002:** WHO category, molecular characteristics, and type of treatment of the study population.

Variable	Total Cohort, *n* = 269	Younger Patients (<60 y), *n* = 123	Older Patients (≥60 y), *n* = 146
**WHO 2016 category, *n*** (**%**)	269	123 (45.7)	146 (54.3)
AML with genetic abnormalities			
t(8;21)	6 (2.2)	5 (4.1)	1 (0.7)
inv16	11 (4.1)	6 (4.9)	5 (3.4)
*KMT2A* rearranged			
t(9;11)	3 (1.1)	2 (1.6)	1 (0.7)
t(11;19)	1 (0.4)	1 (0.8)	0
t(6;9)	2 (0.7)	2 (1.6)	0
inv3	0	0	0
mutated *NPM1*	68 (25.3)	33 (26.8)	35 (24.0)
AML with Myelodysplasia-related changes	54 (20.1)	15 (12.2)	39 (26.7)
Therapy related	27 (10.0)	16 (13.0)	11 (7.5)
AML, NOS	95 (35.3)	43 (35.0)	52 (35.6)
Myeloid sarcoma	2 (0.7)	0	2 (1.4)
**ELN 2017 Risk stratification, *n*** (**%**)			
Favorable	57 (21.2)	31 (25.2)	26 (17.8)
Intermediate	50 (18.6)	29 (23.6)	21 (14.4)
Adverse	98 (36.4)	30 (24.4)	68 (46.6)
Favorable/Intermediate	26 (9.7)	14 (11.4)	12 (8.2)
Intermediate/Adverse	11 (4.1)	7 (5.7)	4 (2.7)
Indeterminate	27 (10.0)	12 (9.8)	15 (10.3)
**Mutated gene, *n*** (**%**)			
*FLT3* *	57 (21.2)	31 (25.2)	26 (17.8)
*FLT3*-ITD	44 (16.4)	25 (20.3)	19 (13.0)
*FLT3*-TKD	15 (5.6)	6 (4.9)	9 (6.2)
*NPM1*	76 (28.3)	37 (30.1)	39 (26.7)
*DNMT3A*	67 (24.9)	28 (22.8)	39 (26.7)
*RUNX1*	25(9.3)	4 (3.3)	21 (14.4)
*ASXL1*	37 (13.8)	10 (8.1)	27 (18.5)
*TP53*	27 (10.0)	5 (4.1)	22 (15.1)
*IDH1*	28 (10.4)	11 (8.9)	17 (11.6)
*IDH2*	36 (13.4)	16 (13.0)	20 (13.7)
*BCOR*	15 (5.6)	7 (5.7)	8 (5.5)
*SETBP1*	3 (1.1)	0	3 (2.1)
*ZRSR2*	6 (2.2)	0	6 (4.1)
*WT1*	12 (4.5)	9 (7.3)	3 (2.1)
**Treatment, *n*** (**%**)			
Intensive chemotherapy	179 (66.5)	114 (92.7)	65 (44.5)
Low-intensive chemotherapy (AZA/LDAC)	38 (14.1)	1 (0.8)	37 (25.3)
Best supportive care	32 (11.9)	2 (1.6)	30 (20.5)
Died before treatment	9 (3.3)	3 (2.4)	6 (4.1)
Not available	11 (4.1)	3 (2.4)	8 (5.5)
HSCT	35 (13.0)	31 (25.2)	4 (2.7)

NOS: not otherwise specified; ELN: European LeukemiaNet; AZA: azacytidine; LDAC: low-dose cytarabine; HSCT: hematopoietic stem cell transplant. * The asterisk highlights that in the indicated group, two patients (≥60 years) had concomitant *FLT3*-ITD and *FLT3*-TKD mutations. Because the *FLT3*-ITD allelic burden [8] was not available, patients containing both *FLT3*-ITD and *NPM1* mutations were considered favorable|intermediate risk, whereas patients containing *FLT3*-ITD but no *NPM1* mutation were considered intermediate|adverse risk.

**Table 3 cancers-14-03236-t003:** Frequency of selected gene mutations according to age.

Mutated Gene, *n* (%)	Total Cohort, *n* = 269	Younger Patients (<60 y), *n* = 123	Older Patients (≥60 y), *n* = 146	*p*-Value
**Group I** *SRSF2*, *SF3B1*, *U2AF1*, *ZRSR2*, *ASXL1*, *EZH2*, *BCOR* or *STAG2*	98 (36)	31 (25)	67 (46)	<0.001 *
**Group II**				
*DNMT3A*	67 (25)	28 (23)	39 (27)	0.456
*ZRSR2*	6 (2)	0	6 (4)	0.033*
*NMP1* and *WT1*	2 (1)	0	2 (1)	0.502
*BCOR*	16 (6)	8 (7)	8 (5)	0.723
*SETBP1*	3 (1)	0	3 (2)	0.253
*IDH2*	36 (14)	16 (13)	20 (14)	0.868
**Group III**				
*ASXL1*	37 (14)	10 (8)	27 (18)	0.014 *
*RUNX1*	25 (9)	4 (3)	21 (14)	0.002 *
*TP53*	27 (10)	5 (4)	22 (15)	0.003 *

Group I includes mutations in at least one myelodysplastic-related gene. Group II includes gene mutations proposed as a refinement of the ELN-2017 criteria. Group III includes ELN-2017 high-risk mutations, excluding *FLT3*-ITD due to lack of information on the allelic burden. Asterisks denote statistically significant differences between younger and older patients.

**Table 4 cancers-14-03236-t004:** Overall survival analysis of patients with selected gene mutations.

Mutated Gene	HR Adjusted	95%CI	*p*-Value
*SRSF2*, *SF3B1*, *U2AF1*, *ZRSR2*, *ASXL1*, *EZH2*, *BCOR*, or *STAG2*	0.70	0.49–1.00	0.051
*DNMT3A*	0.85	0.56–1.28	0.427
*ZRSR2*	1.70	0.62–4.68	0.304
*NMP1* and *WT1*	1.48	0.36–6.08	0.587
*BCOR*	0.76	0.38–1.51	0.431
*SETBP1*	3.87	0.93–16.02	0.062
*IDH2*	0.68	0.40–1.17	0.166
*NPM1*	0.96	0.52–1.78	0.908
*TP53*	2.96	1.81–4.84 84	<0.001
*PTPN11*	0.80	0.44–1.45	0.459

The hazard ratio (HR) for each mutation was tested separately adjusting for age at diagnosis (categorized as either <60 or ≥60 years old) and ELN-2017 risk group.

## Data Availability

The data presented in this study are available on request from the corresponding author.

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
