# Peer review of "Screening a Targeted Panel of Genes by Next-Generation Sequencing Improves Risk Stratification in Real World Patients with Acute Myeloid Leukemia"

_cancers, 2022, doi:10.3390/cancers14133236_

Round 1
Reviewer 1 Report
In this study, the investigators analyzed 268 newly diagnosed patients with AML by next generation sequencing. They were able to risk stratify better 62/268 by NGS(23%). They also noted unexpected outcomes associated with FLT3 mutations.
This is a good effort. My greatest concern is lumping young and old, and intensive and low intensity treated patients in the same analysis. This resulted in showing that ELN classification applied poorly to OS, and that mutations had borderline significance. I have the following suggestions:
1) Abstract-- fine. Please clarify what what "unexpected" with FLT3
2) Introduction -- fine
3) Patients and Methods -- fine
4) Results -- Patient median age only 61 years, much younger than expected in AML ( median age 69-72 years). Please explain why. Did you exclude some older patients from NGS analysis. Were these 268 consecutive patients or selected younger patients?
5) Results paragraph 2 -- only 34% of patients had CG abnormalities. Usually it is 50% or even more. Again did you exclude older patients, therapy related AML, AML evolving from MDS/MPN?
6) Results, therapy -- 66% received intensive chemo, 14% low intensity chemo, and 12% best supportive care. The latter is worrisome in a young population. How were patients selected for each of the 3 options?
7) CR+ CRi rate 64%. This is low. What was the CR rate? What was the ORR by therapy?
8) 9+13 patient died in induction out of 179 treated = 12% mortality ( excluding ones who received supportive care). This is quite high for a younger population. Please explain. Did they received antibiotic/antifungal prophylaxis? Was treatment in or outpatient?
9) Need to have Figure 1 as overall survival curve and survival by intensive chemo, low intensity chemo, and no therapy
10) Results, p7 -- 44 patients were FLT3 ITD by PCR; only 21 detected by NGS. This is known and very important. How many of the 44 received FLT3 inhibitors? Outcome ( CR;OS)?
11) Results, p5 -- only 35 patients ( 13%) had allo SCT. Why such a low %?
11) pages 6-9 -- Analyses of mutational landscape and co-occurrences well known from literature. No surprises
12) Figure 3 -- It seems OS not as expected. INT-ADV did best, FAV and INT did the same, while FAV-INT did worse. Can you please explain? Maybe need Figure 3A as is, 3B for intensive chemo, 3C for low intensity chemo. The ELN applies well to younger patients on intensive chemo and poorly to older/unfit and low intensity chemo. This is a very important point to highlight. Again we need a Figure for OS of all 268 patient, and include the 3 treatment subsets.
13) In fact, the best and novel contribution of this paper might be, not the ELN reclassification, but that the ELN applies well to younger/fit on intensive chemo and poorly to older/unfit on low intensity therapy
14) I think doing the MVA on the total population, not accounting for age and CG gives a wrong picture of the importance of mutations. It is now clear that once age and CG are factored, only NPM1, TP53 and PTPN11 are important. FLT3 ITD may or may not be depending on how much FLT3 inhibitors and allo SCT were used in this setting. Table 4 actually shows that NONE of the mutations are relevant.
15) Discussion needs to be re-written AFTER above points considered. I think this study emphasized only the reclassification by NGS. But doing the overall analysis including all ages and therapies resulted in showing that ELN applies poorly to OS, and that mutations were not important
Reviewer 2 Report
In the manuscript „Screening a targeted panel of genes by next generation sequencing improves risk stratification in real world patients with Acute Myeloid Leukemia“ authors analyzed the cohort of 268 de novo AML patients and performed the NGS panel targeting full exonic regions of 15 genes and exonic hotspots of additional 39 genes. First, the descriptive statistics on demographic and laboratory characteristics is presented, followed by WHO category, molecular characteristics and type of treatment of the study population, mutational landscape and mutational landscape and frequency according to age. Finally, the authors report on the survival according to the age category and ELN-2017 risk group, compare patients stratification with and without NGS and additionally the contribution of NGS to OS of AML patients.
Broad comments: The manuscript is decently written and reports on the important matter but there is not much novelty, and the interesting findings that the authors noticed have not been properly discussed.
Specific comments:
1. The authors discuss their results that confirmed the previous investigations (frequency of genes, dismal outcome due to the mutation of some genes...). Please highlight the novelty of your study.
2. Table 2 – NPM1 28.3%, line 245 – NPM1 28.2%, please check all the numbers.
3. Please discuss in more details the reason why in the favorable/intermediate group the risk was less favorable than in the intermediate group– eg. have you noticed that favorable/intermediate group has statistically significantly more DNMT3A and IDH1/2 mutations than intermediate group? Are there any other mutations that differ and could be important regarding this finding The figure with that data could be helpful.
4. The authors just marginally state that there are almost half of patients in intermediate/adverse group with DNMT3 mutations which had better outcome than in favorable group. Do only these patients with DNMT3 mutation have better outcome? What could be the logics behind this if, as the authors already mentioned, FLT3 with DNMT3 mutation is the reason for the adverse prognostic effect (PMID: 34376373)
5. The authors did not discuss lack of differences in overall survival after the incorporation of NGS data and including more patients in the adverse risk group – was the reason intensive chemotherapy, infections etc..?
Reviewer 3 Report
The article of Matos et al. reports results of a real-world study of newly diagnosed AML patients and evaluated interest of parallel NGS testing for myeloid genes compared to PCR-testing of NPM1/FLT3 etc. Interestingly,62 patients could be reclassified due to NGS data otherwise these patients were either misclassified or failed risk classification according to the ELN-2017 recommendations. These real world findings are of high interest especially for the validation of routine use of NGS for AML patients to justify reimbursement of health insurances.
Overall the paper is very well written and adds value to the field.
Minor remarks:
1.) The findings for FLT3-ITD patients are intriguing in particular outcome for FLT3-ITD+ patients without other mutations associated. Nevertheless, FLT3-ITD allele ratio determination is difficult and NGS is not validated for this purpose. Authors should stress a little bit more this point in the discussion.
2.) The patient population is limited. If possible a second real life control cohort would be of interest. That should at least be discussed.
Round 2
Reviewer 1 Report
Accept
Author Response
We thank the reviewer for accepting our corrections.
We re-checked the text and misspellings were corrected.
Reviewer 2 Report
The authors have made some changes to the manuscript but they have unfortunately disregarded nearly all my questions raised, and because of the clinical. importance of the study, the majority of them will be again repeated, hoping for the appropriate discussion on these matters.
1. Figure 4 is all mixed up – I can see only four panels and there are obviously some panels underneath (a,b,c,d,e,f)
2. I do not see a paragraph (both in results and discussion section) concerning question raised previously, especially is there a statistical difference between groups regarding gene mutations: why in the favorable/intermediate group the risk was less favorable than in the intermediate group– eg. have you noticed that favorable/intermediate group has statistically significantly more DNMT3A and IDH1/2 mutations than intermediate group? The authors state “Further analysis revealed that the majority (96%) of these patients had mutations in additional relevant genes, including DNMT3A and IDH1/2.“ Is this statistically significant in comparison to the intermediate group? If yes, could that be the reason behind more adverse prognosis?
Are there any other mutations that differ and could be important regarding this finding. The figure with that data could be helpful.
3. Authors did not discuss the question previously raised: Do only these patients in intermediate/adverse group with DNMT3 mutation have better outcome? What could be the logics behind this if, as the authors already mentioned, FLT3 with DNMT3 mutation is the reason for the adverse prognostic effect (PMID: 34376373)
4. The authors showed the survival of for intensive treatment, supposedly that corresponds to adverse risk group and show that there is an increase in OS. Is that regardless of inclusion of NGS data? Again - Figure 5 - The authors did not discuss lack of differences in overall survival after the incorporation of NGS data and including more patients in the adverse risk group – was the reason intensive chemotherapy, infections etc..?
Author Response
"Please see the attachment."

Round 3
Reviewer 2 Report
The authors have explained all the questions raised and I have no further suggestions.